# Effectiveness and Safety of Linezolid Versus Vancomycin, Teicoplanin, or Daptomycin against Methicillin-Resistant *Staphylococcus aureus* Bacteremia: A Systematic Review and Meta-Analysis

**DOI:** 10.3390/antibiotics12040697

**Published:** 2023-04-02

**Authors:** Hitoshi Kawasuji, Kentaro Nagaoka, Yasuhiro Tsuji, Kou Kimoto, Yusuke Takegoshi, Makito Kaneda, Yushi Murai, Haruka Karaushi, Kotaro Mitsutake, Yoshihiro Yamamoto

**Affiliations:** 1Department of Clinical Infectious Diseases, Toyama University Graduate School of Medicine and Pharmaceutical Sciences, Toyama 930-0194, Japan; 2Laboratory of Clinical Pharmacometrics, School of Pharmacy, Nihon University, Chiba 274-8555, Japan; 3Department of Infectious Diseases and Infection Control, International Medical Center, Saitama Medical University School of Medicine, Saitama 350-1298, Japan

**Keywords:** meta-analysis, linezolid, effectiveness, methicillin-resistant *Staphylococcus aureus*, bacteremia

## Abstract

Vancomycin (VCM) and daptomycin (DAP) are standard therapies for methicillin-resistant *Staphylococcus aureus* (MRSA) bacteremia, despite concerns regarding clinical utility and growing resistance. Linezolid (LZD) affords superior tissue penetration to VCM or DAP and has been successfully used as salvage therapy for persistent MRSA bacteremia, indicating its utility as a first-choice drug against MRSA bacteremia. In a systematic review and meta-analysis, we compared the effectiveness and safety of LZD with VCM, teicoplanin (TEIC), or DAP in patients with MRSA bacteremia. We evaluated all-cause mortality as the primary effectiveness outcome, clinical and microbiological cure, hospital length of stay, recurrence, and 90-day readmission rates as secondary effectiveness outcomes, and drug-related adverse effects as primary safety outcomes. We identified 5328 patients across 2 randomized controlled trials (RCTs), 1 pooled analysis of 5 RCTs, 1 subgroup analysis (1 RCT), and 5 case-control and cohort studies (CSs). Primary and secondary effectiveness outcomes were comparable between patients treated with LZD versus VCM, TEIC, or DAP in RCT-based studies and CSs. There was no difference in adverse event incidence between LZD and comparators. These findings suggest that LZD could be a potential first-line drug against MRSA bacteremia as well as VCM or DAP.

## 1. Introduction

Methicillin-resistant *Staphylococcus aureus* (MRSA) bacteremia has been persistently associated with a mortality rate exceeding 20% despite standardization of management and improved quality of care based on established guidelines and reviews [1,2]. Although it is evident that selecting based on the source, optimal dosing, and timing of antibacterial therapy and source control can markedly impact treatment outcomes in MRSA bacteremia [2,3], with limited treatment options and persistent high mortality, we need to consider the wider applicability of off-labeled drugs to MRSA bacteremia.

Vancomycin (VCM) and daptomycin (DAP) have long been considered standard therapies for MRSA bacteremia [4]. Linezolid (LZD), an oxazolidinone antibiotic, has been studied off-label for *S. aureus* bacteremia [5,6,7,8,9]; however, data remain limited, and its delayed application to treat bacteremia can be attributed to bacteriostatic effect, drug–drug interactions, and the increased risk of myelosuppression during prolonged treatment [10,11]. Unfortunately, the dogma that bactericidal antibiotics afford greater clinical effectiveness than bacteriostatic agents persists; however, a recent systematic review has revealed no substantial difference in effectiveness when comparing bacteriostatic and bactericidal agents for serious infections, thereby suggesting that drug characteristics such as pharmacokinetics, tissue distribution, and penetration are more relevant than arbitrary laboratory terminologies [12].

LZD offers the option for oral treatment with nearly 100% bioavailability and excellent tissue penetration when compared with that of VCM or DAP [13]. Recent systematic reviews and meta-analyses found that clinical cure and microbiologic success in proven MRSA pneumonia or complicated skin and soft tissue infections (cSSTIs) were superior in patients treated with LZD compared with patients treated with VCM, without increasing the risk of drug-related adverse events [14,15]. With regard to bacteremia, cases of persistent MRSA bacteremia have been successfully treated with LZD as salvage therapy [16,17]. Although data are limited, LZD could be a potential first-line drug for MRSA bacteremia, as well as pneumonia, cSSTIs, or central nervous system infections, to reduce the unbalanced use of anti-MRSA agents or halt the further progression of drug resistance.

To date, several studies have investigated and compared the effectiveness and safety of LZD with VCM, teicoplanin (TEIC), and DAP in the context of MRSA bacteremia [5,6,8,9,18]. More recently, the updated UK MRSA treatment guidelines recommend LZD as an alternative first-line drug to treat MRSA bacteremia when VCM is contraindicated (strong recommendation) and proposed TEIC and DAP as alternatives when VCM and LZD are contraindicated (weak recommendation) [19]. However, no notable evidence supporting these recommendations has been presented, and to the best of our knowledge, no meta-analytic comparison has been reported. Consequently, the objective of the present systematic review and meta-analysis was to comprehensively evaluate the effectiveness and safety of LZD versus VCM, TEIC, and DAP for treating MRSA bacteremia. We hypothesized that LZD would afford similar effectiveness and safety to VCM, TEIC, or DAP in terms of mortality rate, clinical and microbiological cure rates, and the incidence rate of drug-related adverse effects.

## 2. Results

### 2.1. Literature Search Results

#### 2.1.1. Systematic Review

A systematic review of electronic databases identified 3121 articles. After reviewing the titles and abstracts, 3074 articles were deemed ineligible. A full-text review of 47 articles was conducted. The literature selection process is illustrated in Figure 1. Subsequently, a total of nine studies met the inclusion criteria [3,6,7,8,9,18,20,21,22], including two randomized controlled trials (RCTs) [7,20], one pooled analysis of five RCTs [6], one subgroup analysis of one RCT [21], and five case-control and cohort studies (CSs) [3,8,9,18,22]. The characteristics of the studies included in the meta-analysis are presented in Table 1. Five studies were conducted in Asia [6,7,20,21,22], six in North America [6,7,8,9,20,21], and six in Western countries [3,6,7,18,20,21]. The included studies comprised 5328 patients, including 75 patients in two RCTs, 73 in one pooled analysis of RCTs, 56 in one subgroup analysis of one RCT, and 5124 patients in five CSs. The sample size of each study ranged between 28 and 4580 patients. In the RCT-based studies (two RCTs, one pooled analysis of five RCTs, and one subgroup analysis of one RCT) [6,7,20,21], 111 patients received VCM and 93 received LZD, with no patients receiving TEIC or DAP. In CSs [3,8,9,18,22], 4783, 27, 114, and 200 patients were treated with VCM, TEIC, DAP, and LZD, respectively. Six studies described VCM dose adjustment or trough concentrations [6,7,8,20,21,22]; however, no study described the initial loading dose or area under the time-concentration curve (AUC)-guided dosing. In addition, no included study described the optimal TEIC doses.

#### 2.1.2. Meta-Analysis

##### Primary Effectiveness Outcomes

The all-cause mortality rates extracted from two RCT-based studies (one pooled analysis of five RCTs [6] and one subgroup analysis of one RCT [21]) were 36.2% (21/58) for patients receiving LZD and 36.6% (26/71) for patients receiving VCM. Mortality showed no significant difference between LZD and VCM (Odds ratio [OR] 1.00, 95% confidence intervals [CI]:0.49–2.07, I^2^ = 0%, Figure 2A). Similarly, the mortality rates in five CSs [3,8,9,18,22] were comparable between LZD and VCM, TEIC, or DAP (OR 0.81, 95% CI:0.30–2.18, I^2^ = 77%, Figure 2B).

##### Secondary Effectiveness Outcomes

There were no significant differences in the clinical cure rates between LZD and VCM or DAP, considering the three RCTs (OR 1.43, 95% CI:0.67–3.06, I^2^ = 0%, Figure 3A) [6,7,20] and one CS (OR 1.01, 95% CI:0.33–3.04, Figure 3B) [8]. In the microbiological cure rate, no significant difference was observed between patients treated with LZD and those treated with VCM, TEIC, or DAP in one RCT (OR 0.70, 95% CI:0.15–3.34, Figure 4A) [7] and two CSs (OR 3.33, 95% CI:0.84–13.14, I^2^ = 0%, Figure 4B) [3,8].

We further examined hospital length of stay (LOS), recurrence rate, and 90-day readmission rate for LZD- versus VCM-, TEIC-, or DAP-treated patients with MRSA bacteremia. Considering hospital LOS, given that Usery et al. [8] reported only the mean and standard deviation (SD) values for the VCM and DAP groups, respectively, we incorporated the calculated data of the integrated VCM and DAP groups. In addition, although Yeager et al. [9] reported only the median and interquartile range of hospital LOS, the corresponding author kindly provided the mean and SD values after a request via e-mail, and we could incorporate relevant data into the meta-analysis. Subsequently, we extracted data on hospital LOS from three CSs [3,8,9]; we found no significant difference between LZD and VCM, TEIC, or DAP (standard mean difference −0.12, 95% CI: −0.45, 0.20, I^2^ = 38%, Figure 5). In addition, we detected no significant differences between LZD and VCM or DAP for the recurrence rates in the two CSs (OR 0.83, 95% CI: 0.35–1.83, I^2^ = 0%, Figure 6A) [8,9] and the 90-day readmission rates in the two CSs (OR 0.81, 95% CI:0.50–1.33, I^2^ = 19%, Figure 6B) [9,18].

##### Primary Safety Outcome

Unlike the meta-analysis of clinical effectiveness, we extracted data on adverse events from only one CS. Considering the overall incidence of drug-related adverse events, there was no significant difference between LZD and VCM or DAP (OR 1.09, 95% CI:0.47–2.50) [9].

### 2.2. Risk of Bias

Appendix A show the risk of bias assessment for all included studies. The risk of bias was low in both RCT-based studies and CSs. However, the risks of bias due to allocation concealment (selection bias), blinding of participants and personnel (performance bias), and blinding of outcome assessment (detection bias) were unclear or high in 50% (two of four) of the included RCT-based studies. Moreover, the risk of bias due to confounding variables (selection bias) and measurement of exposure (performance bias) was unclear or high risk in 80% (four of five) of included CSs.

### 2.3. Quality of Evidence

The present meta-analysis comprised four RCT-based studies, including a pooled analysis of five RCTs and a subgroup analysis of one RCT. Given the inclusion of five CSs, the risk of bias, indirectness, and imprecision, the final grade of the evidence was evaluated as low (⊕⊕⊖⊖), indicating that our confidence in the effect estimate was limited.

## 3. Discussion

To the best of our knowledge, the present study is the first meta-analysis to evaluate the effectiveness and safety of LZD for MRSA bacteremia when compared with that of VCM, TEIC, or DAP. Although TEIC is unavailable in the US, it has been approved in Asia and Europe for use in bacteremia associated with several Gram-positive infections and is now widely used as an effective and safe alternative to vancomycin in treating healthcare-associated MRSA bacteremia [2,26]; hence, we included TEIC as a comparator. Consequently, the present meta-analysis, including four RCT-based studies and five CSs, demonstrated that the incidence rates of mortality, clinical and microbiological cure, recurrence, 90-day readmission, drug-related adverse events, and hospital LOS did not significantly differ between LZD- and VCM-, TEIC-, or DAP-treated with MRSA bacteremia.

Given that linezolid is the only agent with an oral formulation and available for outpatient therapy, shorter hospital LOS in the patients treated with LZD was anticipated. However, in the present meta-analysis, a significant difference was not observed between LZD and VCM, TEIC, or DAP, which is consistent with the result in meta-analysis of LZD and DAP for the treatment of VCM-resistant enterococcal bacteremia [27]. On the other hand, for SSTIs, the previous systematic review and meta-analysis showed that the median hospital LOS was three days shorter in the patients treated with LZD than those with VCM [28]. This discrepancy would be due to the differences in the source or severity of the infection.

VCM and DAP have long been first-line antibiotics for MRSA bacteremia treatment [4]; however, concerns regarding their clinical utility and gradually increasing resistance persist. VCM is the most commonly used, although it has a relatively slow onset of bactericidal activity and poor penetration of certain tissues [2]. TEIC, like VCM, also has a slow bactericidal action, and in vitro time-kill analysis showed that TEIC required 24 h to completely eliminate MRSA, even at high concentrations [29]. In addition, MRSA with a reduced vancomycin susceptibility phenotype, including heterogeneous vancomycin-intermediate *S. aureus* (hVISA) and MRSA with elevated VCM minimum inhibitory concentration (MIC; ≥ 1.5 mg/L) have been increasingly reported over the past decade, potentially attributed to the unilateral use of VCM [30], which further clarifies the recent worldwide spread of VCM-resistant Enterococci [31].

Observational studies over the past several years have demonstrated that MRSA infections with elevated VCM MIC were associated with substantially higher treatment failure and poor outcomes when treated with VCM [30]. DAP remains an alternative treatment option in these cases; however, MICs for VCM and DAP are correlated [32], and up to 15% of hVISA isolates exhibit additional non-susceptibility to DAP [33]. Furthermore, prior vancomycin failure has been correlated with the acquisition of heteroresistance and reduced success of DAP therapy [2,34].

LZD has favorable pharmacokinetic properties, including availability in both intravenous and oral formulations, high oral bioavailability (approximately 100%) [35], and excellent penetration into the lungs [36,37], skin [38,39], muscles [40,41], bones [42,43,44], and cerebrospinal fluid [45,46,47] when compared with those of VCM or DAP. Therefore, in clinical settings, superiority or non-inferiority to other anti-MRSA agents has been confirmed in several types of infections, including pneumonia [14], cSSTIs [15], and central nervous system infections [48] caused by MRSA, resulting in national guidelines that recommend LZD as the first-line or alternative drug for treating above-mentioned infections [4,19].

Considering bacteremia, the US Food and Drug Administration (FDA) warned against the use of LZD for treating catheter-associated bloodstream infections (BSIs) in 2007; this prevented the conduction of high-quality studies and further minimized the use of LZD in bacteremia [49]. However, this warning was based on a study that reported LZD was associated with an increased risk of death in a subgroup of patients with Gram-negative BSIs, whereas no increased mortality was observed in those with Gram-positive BSIs [7]. Based on the findings of the present meta-analysis, we propose that LZD could be one of the first-line drugs for MRSA bacteremia. Moreover, our evidence supports the recommendations of the updated UK guidelines [19].

On the other hand, although a recent study showed that resistance to LZD in staphylococci is low [50], first-line use of linezolid for MRSA bacteremia should be discussed carefully because it may lead to the development of LZD resistance and restrict the treatment options for staphylococci resistant to glycopeptides.

Drug-related adverse events, including nephrotoxicity and thrombocytopenia, did not significantly differ between LZD and VCM or DAP; however, only one study was included for adverse events assessment, which may not be representative and lacks power. In addition, LZD has several serious adverse effects, particularly in long-term use, such as thrombocytopenia and neuropathy, which leads to treatment discontinuation and failure. Therefore, the severity of side effects should be assessed, as well as the number. The previous meta-analysis of 9 RCTs, involving 5249 patients, to compare the efficacy and safety of LZD with VCM for MRSA-related infections, including but not limited to bacteremia, pneumonia, cSSTIs, and urinary tract infections, showed that the linezolid therapy group was associated with significantly more gastrointestinal-related adverse events and significantly fewer episodes of abnormal renal function [51]. On the other hand, there were no differences regarding the overall incidence of drug-related adverse events and serious adverse events and similar episodes of thrombocytopenia and anemia between the LZD and the VCM groups [51]. However, it is well-established that LZD can cause myelosuppression; hence, complete blood cell counts should be monitored during LZD therapy. Moreover, surveillance for adverse events should be increased in the presence of other risk factors for myelosuppression, such as age, low body weight, renal impairment, or prolonged therapy [52,53,54,55].

The present study has several limitations. First, two of the four RCT-based studies were conducted in patients with *S. aureus* bacteremia and the others in those with MRSA pneumonia. The patients with MRSA bacteremia may not have been equally allocated, and the risk of bias due to randomization or allocation was not all low risk; thus, these biases may have influenced the results. In addition, three of the five CSs selected for the meta-analysis were retrospective, with only one adjusting for potential confounding factors using propensity score matching. A significant number of patients in the LZD, TEIC, or DAP groups were empirically switched from VCM, which may have influenced the results. Second, we did not prospectively register the study using a systematic review protocol in an international database. However, given that the present study was conducted in accordance with PRISMA2020 (Appendix A) [56], we consider the results of the study to be valid. Third, our meta-analysis did not include studies that assessed the effectiveness and safety of VCM with loading doses and AUC-guided monitoring according to the latest recommendations [57,58]. A VCM loading dose is essential for achieving an early target concentration, and AUC-guided dosing was found to be independently associated with reduced nephrotoxicity [57,58]. Fourth, patients with MRSA bacteremia tend to have complications including infective endocarditis, orthopedic infections (e.g., septic arthritis and osteomyelitis), abscesses, and prosthetic device-related infections, which often need non-pharmacological intervention such as surgery, drainage, and device removal. These factors may have markedly impacted the observed results, but we could not assess their influence on outcomes. Fifth, we could not perform stratified analyses of disease severity, primary source, or vancomycin trough levels, given that prognostic factors such as individual data were unavailable. Finally, there is a significant lack of data regarding the safety profile of linezolid and the duration of the therapy. We could not conduct a meta-analysis since there was only one study assessing the adverse effects. Further studies are needed to validate the safety profile of linezolid for MRSA bacteremia.

In conclusion, the present study revealed that LZD was comparable to VCM, TEIC, and DAP as a definitive treatment for MRSA bacteremia in terms of clinical effectiveness. Although additional high-quality studies are needed to confirm this observation, our findings provide robust evidence to corroborate the previously reported assertion of LZD non-inferiority over VCM or DAP. Thus, we recommend LZD as a first-choice drug against MRSA bacteremia.

## 4. Materials and Methods

### 4.1. Search Strategy and Study Selection Criteria

This study was conducted in accordance with the PRISMA guidelines for reporting systematic reviews and meta-analyses [56]. The following PICO criteria were used for study selection: patient population (P), adult patients with proven MRSA bacteremia; intervention (I), LZD; comparison (C), VCM, TEIC, or DAP; (O), all-cause mortality, clinical cure, microbiological cure, LOS, recurrence, 90-day readmission, and drug-related adverse effects (new-onset thrombocytopenia, acute kidney injury, and creatine phosphokinase [CPK] elevation). We performed a literature search in the following four electronic databases: PubMed, Web of Science, Cochrane Library, and ClinicalTrials.gov on 16 January 2023. Four authors (HiK, KN, YaT, and KK) independently searched the literature using the terms listed in Appendix A. Duplicate articles were excluded. If the original publication did not include sufficient information regarding outcomes, we requested additional data from the corresponding authors via e-mail. Despite requests, studies were excluded if they did not provide sufficient data on MRSA bacteremia or each antibiotic agent to be included in the meta-analysis.

### 4.2. Data Extraction

Data including authors, publication year, study design, period, country, patient age, the total number of patients, dose regimens of each antibiotic agent, treatment duration, and clinical outcomes were extracted from all studies. We analyzed all-cause mortality as the primary effectiveness outcome. In addition, clinical and microbiological cure rates, hospital LOS, recurrence rates, and 90-day readmission rates were analyzed as secondary effectiveness outcomes. As the primary safety outcome, we evaluated the overall incidence of drug-related adverse effects, including thrombocytopenia (defined as a decrease in platelet count by ≥50% of baseline or platelet count <150 × 10^3^/μL during or following antibiotic treatment), acute kidney injury (defined as an increase in serum creatinine ≥0.3 mg/dL within 48 h or an increase in serum creatinine ≥1.5 the baseline within the prior 7 days), and CPK elevation (CPK >5 times the upper limit of normal [ULN] or 1000 U/L in the presence of myopathy symptoms, or CPK ≥10 times ULN or 2000 U/L without myopathy symptoms during/after treatment with daptomycin) [9].

### 4.3. Assessment for the Risk of Bias

We assessed the risk of bias in RCT-based studies using the Cochrane risk-of-bias tool [59]. The risk of bias in CSs was assessed using the Risk of Bias Assessment Tool for Nonrandomized Studies (RoBANS) [60]. Three reviewers (HiK, KN, and YaT) independently assessed the risk of bias and resolved all discrepancies by consensus in all included studies.

### 4.4. Assessment of Quality of Evidence

We assessed the quality of the evidence using the Grading of Recommendations Assessment Development and Evaluation (GRADE) guidelines [61]. GRADE specifies that the quality of the evidence can be classified into four categories according to the corresponding evaluation criteria: (1) high (⊕⊕⊕⊕); (2) moderate (⊕⊕⊕⊖); (3) low (⊕⊕⊖⊖); and (4) very low (⊕⊖⊖⊖).

### 4.5. Statistical Analysis

The statistical analysis was performed using Review Manager for Mac (RevMan, Version 5.4, The Nordic Cochrane Center, Copenhagen, Denmark), and forest plots were generated. ORs and 95% CIs were calculated using the Mantel–Haenszel method and a random effects model. Statistical heterogeneity among studies was assessed using the I^2^ statistic. I^2^ values ≥ 50%, 25–50%, and ≤25% were regarded as strong, moderate, and no heterogeneity, respectively.

## Figures and Tables

**Figure 1 antibiotics-12-00697-f001:**
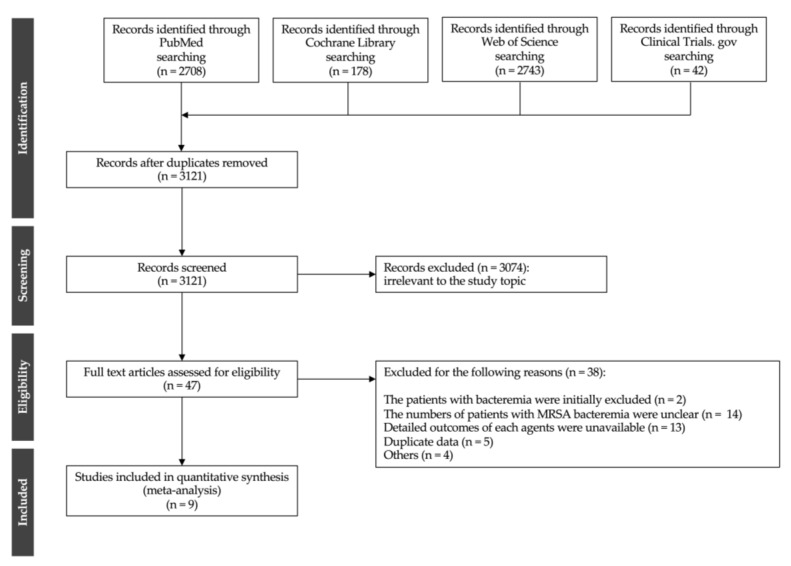
Flow chart of the study selection process.

**Figure 2 antibiotics-12-00697-f002:**
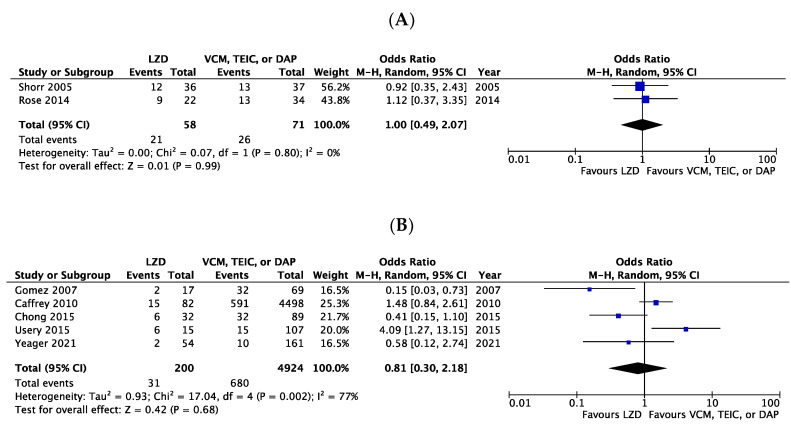
Forest plots of all-cause mortality in patients with MRSA bacteremia treated with LZD versus VCM, TEIC, or DAP. (**A**) RCTs; (**B**) CSs. The vertical line indicates no significant difference between compared groups. Mantel–Haenszel ORs are represented by diamond shapes, and 95% CIs are represented by horizontal lines. Squares indicate point estimates, and the square size indicates the weight of each study included in the meta-analysis. MRSA, methicillin-resistant *Staphylococcus aureus*; RCTs, randomized controlled trial; CSs, case-control and cohort studies; CI, confidence interval; LZD, linezolid; VCM, vancomycin; TEIC, teicoplanin; DAP, daptomycin; OR, odds ratio.

**Figure 3 antibiotics-12-00697-f003:**
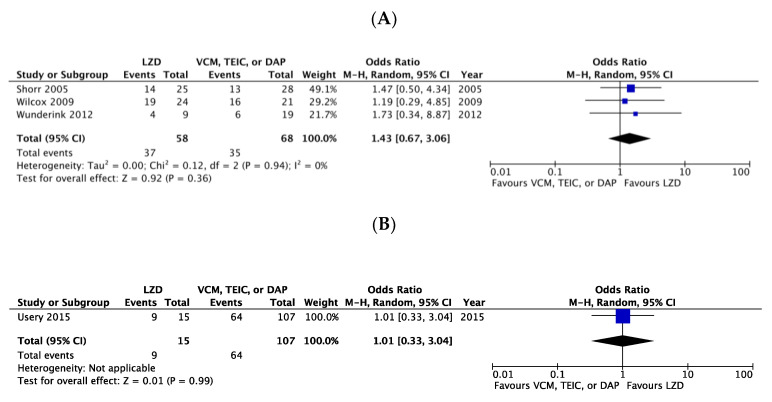
Forest plots of clinical cure rates in patients with MRSA bacteremia treated with LZD versus VCM, TEIC, or DAP. (**A**) RCTs; (**B**) CSs. The vertical line indicates no significant difference between the compared groups. Mantel–Haenszel ORs are represented by diamond shapes, and 95% CIs are represented by horizontal lines. Squares indicate point estimates, and the square size indicates the weight of each study included in the meta-analysis. MRSA, methicillin-resistant *Staphylococcus aureus*; RCTs, randomized controlled trial; CSs, case-control and cohort studies; CI, confidence interval; LZD, linezolid; VCM, vancomycin; TEIC, teicoplanin; DAP, daptomycin; OR, odds ratio.

**Figure 4 antibiotics-12-00697-f004:**
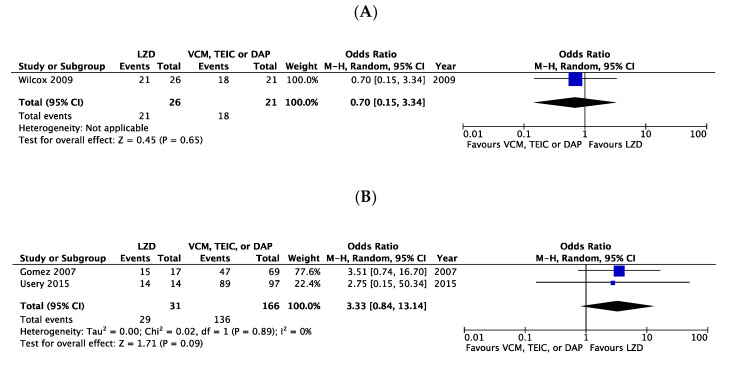
Forest plots of microbiological cure rates in patients with MRSA bacteremia treated with LZD versus VCM, TEIC, or DAP. (**A**) RCTs; (**B**) CSs. The vertical line indicates no significant difference between compared groups. Mantel–Haenszel ORs are represented by diamond shapes, and 95% CIs are represented by horizontal lines. Squares indicate point estimates, and the square size indicates the weight of each study included in the meta-analysis. MRSA, methicillin-resistant *Staphylococcus aureus*; RCTs, randomized controlled trial; CSs, case-control and cohort studies; CI, confidence interval; LZD, linezolid; VCM, vancomycin; TEIC, teicoplanin; DAP, daptomycin; OR, odds ratio.

**Figure 5 antibiotics-12-00697-f005:**
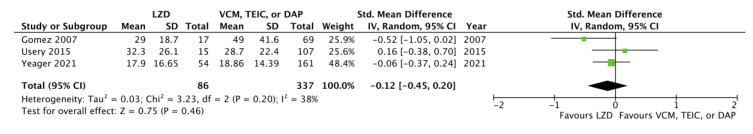
Forest plots of hospital length of stay in patients with MRSA bacteremia treated with LZD versus VCM, TEIC, or DAP. The vertical line indicates no significant difference between the groups compared. The standardized mean differences are represented by diamond shapes, and 95% CIs are represented by horizontal lines. Squares indicate point estimates, and the square size indicates the weight of each study included in the meta-analysis. MRSA, methicillin-resistant *Staphylococcus aureus*; CI, confidence interval; LZD, linezolid; VCM, vancomycin; TEIC, teicoplanin; DAP, daptomycin.

**Figure 6 antibiotics-12-00697-f006:**
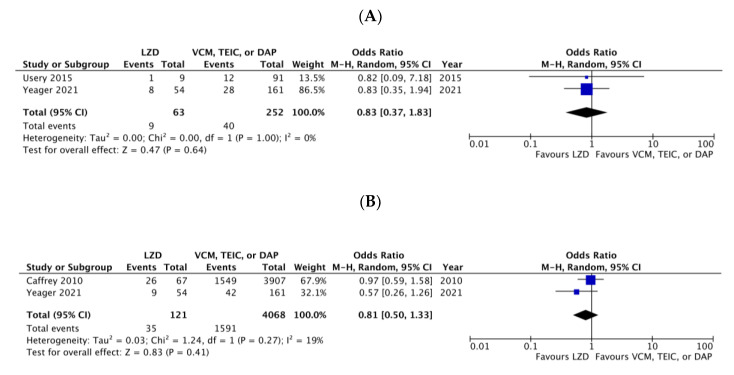
Forest plots of recurrence and 90-day readmission rates in patients with MRSA bacteremia treated with LZD versus VCM, TEIC, or DAP. (**A**) recurrence rates; (**B**) 90-day readmission rates. The vertical line indicates no significant difference between the groups compared. Mantel–Haenszel ORs are represented by diamond shapes, and 95% CIs are represented by horizontal lines. Squares indicate point estimates, and the square size indicates the weight of each study included in the meta-analysis. MRSA, methicillin-resistant *Staphylococcus aureus*; CI, confidence interval; LZD, linezolid; VCM, vancomycin; TEIC, teicoplanin; DAP, daptomycin.

**Table 1 antibiotics-12-00697-t001:** Characteristics of studies included in the meta-analysis.

Study	Study Design	Study Period	Country	Age (Years)	No. of Eligible Patients		Drug Regimen				VCM trough (mg/L)	Treatment Duration (days)	Bacteremia Source	Outcomes
					LZD	VCM, TEIC, or DAP	LZD	VCM	TEIC	DAP				
Shorr 2005 [6]	Pooled analysis of five RCTs [23,24,25]	July 1998 to March 2003	North and South America, Latin America, Europe, Israel, South Africa, Australia, and Asia (MC)	≥13	36	VCM 37	600 mg every 12 h iv, po	1 g every 12 h iv ^a^	-	-	NR	NR	Pneumonia [23,24], SSTI [25], UTI [25], other ^d^ [25]	MO, CC
Gómez 2007 [3]	Prospective observational study	January 2000 to December 2014	Spain (SC)	Mean (range), LZD: 66 (15–95), VCM and TEIC: 58 (14–90)	17	VCM 49, TEIC 20	NR	NR	NR	-	NR	NR	Overall (both LZD, VCM, and TEIC groups): venous catheters (40%), cutaneous (27%), unclear (28%), other (respiratory and urinary) (5%)	MO, MC, LOS
Wilcox 2009 [7]	RCT	May 2002 to August 2005	Argentina, Australia, Austria, Belgium, Brazil, Chile, Colombia, Czech Republic, Germany, Greece, Guatemala, Hungary, India, Italy, Mexico, Pakistan, Peru, Philippines, Russian Federation, Slovakia, South Africa, Spain, Thailand, Turkey, United States, and Venezuela (MC)	≥13	26	VCM 21	600 mg every 12 h iv, po	1 g every 12 h iv ^b^	-	-	NR	NR	CRBSI	CC, MC
Caffrey 2010 [18]	Retrospective cohort study	January 2002 to June 2008	Iceland (MC)	≥18	82	VCM 4498	NR	NR	-	-	NR	NR	NR	MO, 90dRA
Wunderink 2012 [20]	RCT	October 2004 to January 2010	United States, Europe, Asia, South America, and other (MC)	≥18	9	VCM 19	600 mg every 12 h iv	15 mg/kg every 12 h ^c^	-	-	Overall (not only bacteremia): median 12.3 (Day 3), 14.7 (Day 6), 16.1 (Day 9)	Overall (not only bacteremia): median 10 days in both LZD and VCM groups	Pneumonia	CC
Rose 2014 [21]	Subgroup analysis of one RCT [20]	October 2004 to January 2010	United States, Europe, Asia, South America, and other (MC)	≥18	22	VCM 34	600 mg every 12 h iv	15 mg/kg every 12 h ^c^	-	-	Overall (not only bacteremia): median 12.3 (Day 3), 14.7 (Day 6), 16.1 (Day 9)	Overall (not only bacteremia): median 10 days in both LZD and VCM groups	Pneumonia	MO
Chong 2015 [22]	Prospective observational cohort study	August 2008 to April 2011	Republic of Korea (SC)	Adult	32	VCM 82, TEIC 7	NR	NR	NR	-	<15 mg/L 33/94 (35.1%)	NR	LZD: metastatic infection 10 (31.3%), IE 6 (18.8%), CRBSI 17 (53.1%), pneumonia 3 (9.4%), SSTI 1 (3.1%), BJI 3 (9.4%), POWI 1 (3.1%), PB 2 (6.3%); VCM and TEIC: metastatic infection 16 (18.0%), IE 6 (6.7%), CRBSI 42 (47.2%), pneumonia 8 (9.0%), SSTI 5 (5.6%), BJI 3 (3.4%), POWI 8 (9.0%), PB 11 (12.4%)	MO
Usery 2015 [8]	Retrospective cohort study	June 2008 to November 2010	United States (SC)	Mean ± SD, LZD: 53.2 ± 18, VCM: 58.7 ± 15.5, DAP: 59.5 ± 16.2	15	VCM 54, DAP 53	600 mg every 12 h	Mean ± SD, 13.6 ± 4 mg/kg/dose	-	Mean ± SD, 6.7 ± 1.8 mg/kg/day	> 15 mg/L 26/46 (56.5%)	Mean ± SD, LZD: 10.1 ± 3.2, VCM: 13.6 ± 7.1, DAP: 16.4 ± 9.6	LZD: osteomyelitis 2 (13.3%), IE 0, pneumonia 5 (33.3%), VCM: osteomyelitis 7 (13.0%), IE 6 (11.1%), pneumonia 11 (20.4%), DAP: osteomyelitis 11 (20.8%), IE 6 (11.3%), pneumonia 3 (5.7%)	MO, CC, MC, 90dRA, LOS, recurrence
Yeager 2021 [9]	Retrospective cohort study	January 2011 to December 2019	United States (SC)	Median (IQR), LZD: 56 (46–69), VCM and DAP: 48 (38–65)	54	VCM 100, DAP 61	NR	NR	-	NR	NR	median (IQR) duration of total antibiotic therapy, LZD: 22 (17–35), VCM and DAP: 45 (29–49); definitive therapy, LZD: 14 (10–19), VCM and DAP: 39 (24–45)	LZD: BJI 3 (6%), IE 2 (4%), SSTI 24 (44%), CRBSI 3 (6%), pneumonia 13 (24%), other 6 (11%), multiple 3 (6%), VCM and DAP: BJI 29 (18%), IE 25 (16%), SSTI 50 (31%), CRBSI 22 (14%), pneumonia 12 (7%), other 13 (8%), multiple 10 (6%)	MO, LOS, recurrence, AEs

RCT, randomized controlled trial; SC, single center; MC multicenter, LZD, linezolid; VCM, vancomycin; TEIC, teicoplanin; DAP, daptomycin; SD, standard deviation; IQR, interquartile range; iv, intravenous; po, perorally; NR, not reported; SSTI, skin and soft tissue infection; UTI, urinary tract infection; CRBSI, catheter-related bloodstream infection; IE, infectious endocarditis; BJI, bone and joint infection; POWI, postoperative wound infection; PB, primary bacteremia; MO, mortality; CC, clinical cure; MC, microbiological cure; LOS, length of stay; 90dRA, 90-day readmission; AEs, adverse events ^a^ Vancomycin dosage adjustments were required for patients with renal impairment and permitted for other patients according to the local standard of care. In blinded studies, a research pharmacist or equivalent non-study personnel monitored the vancomycin dosages. ^b^ Vancomycin could be adjusted for renal function according to local practice. ^c^ Pharmacist monitored and adjusted vancomycin doses according to local protocols based on trough levels and renal impairment. ^d^ Other includes catheter-associated infection, intra-abdominal or pelvic infection, laryngotracheobronchitis, mediastinitis, infected device, bacteremia secondary to parotitis, empyema, lumbar fistula, sinusitis, and subgaleal empyema. One patient with right-sided endocarditis was included in this category.

## Data Availability

The data presented in this study are available upon request from the corresponding author.

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
