# Peer review of "Effectiveness and Safety of Linezolid Versus Vancomycin, Teicoplanin, or Daptomycin against Methicillin-Resistant Staphylococcus aureus Bacteremia: A Systematic Review and Meta-Analysis"

_antibiotics, 2023, doi:10.3390/antibiotics12040697_

Round 1

Reviewer 1 Report

This systematic review and meta-analysis provides interesting data on the use of linezolid to treat methicillin-resistant Staphylococcus aureus bacteremia, overcoming the long-standing axiom that bacteriostatic drugs are inferior to bactericidal drugs against these infections. The sound effectiveness and handiness of linezolid - given also its favourable pharmacokinetic profile and high bioavailaibility also in the oral form - make it a good choice as first line therapy: nonetheless, few and fragmented studies have compared linezolid to the standard-of-care so far, and the present study might help filling this gap. 

The English language is plain and fluent. All the selected studies are valid and consistent with the aim of the analysis. The data are clearly presented with the support of a forest plot for each considered outcome. The risk of bias and the quality of evidence are clearly exposed, as well as the study limitations listed in the discussion. The discussion itself is concise and well argued. 

The main observation is the lack of data regarding the safety profile of linezolid (provided only by one study) and the duration of the therapy. Furthermore, as the authors underline, data regarding the total drug monitoring of vancomycin and teicoplanin might be helpful to better assess the non-inferiority of linezolid. 

Reviewer 2 Report

-        In the abstract add Linezolid before the first abbreviation (LZD)

-        The authors indicated in the abstract that Linezolid was not inferior; I think the author should be clearer in his statement such as better/equivalent ...etc

-        In the conclusion of the abstract the authors stated Linezolid is better is this the case? Then state why!

-        More evidence need to be stated in the introduction about the effectiveness of the LZD in the treatment against MERSA

-        In the methodology section line 91-95 is confusing. Please rephrase

-        In the methodology part; the confidence level in the confidence interval should be stated (95%)

-        The author should justify why he did not do metanalysis in studying the reverse effect

-        The author should discuss more detail about the effect of bias due to allocation and randomization which is present in some of the clinical trials on their result

-        General note: Double check the language for some grammatical and typing mistakes

Reviewer 3 Report

The authors performed a meta-analysis to evaluate the effectiveness and safety of linezolid for MRSA bacteramia compared with vancomycin, teicoplanin and daptomycin. It is recommended linezolid is a first-choice drug against MRSA bacteramia. The reviewer considers the present manuscript very good. Just a correction: write linezolid before (LZD) line 22;

Reviewer 4 Report

Linezolid is highly effective against MRSA and penetrates well into tissue. The review of its efficacy is therefore useful. However, it can be the only available option against some staphylococci resistant to glycopeptides. Resistance to linezolid can develop.  It has several serious adverse effects, particularly in long term use, such as thrombocytopenia and neuropathy. The severity of side effects should have been assessed as well as the number. Teicoplanin and vancomycin are very widely used and the adverse effect profiles are well known. Linezolid is used for those failing first line treatment. First line use of linezolid could be damaging in terms of antimicrobial stewardship. These issues should be discussed.

P2 Teicoplanin and vancomycin are not rapidly bactericidal. Teicoplanin acts over 24 hours. The methodology used is appropriate. Risk of bias was checked by independent assessment and GRADE used for quality check.  

The lack of difference in efficacy is expected. However, given that linezolid is the only agent that can be given by mouth, a difference in length of stay would be anticipated. Only one study was used for adverse event assessment, which may not be representative and lacks power. Is there any reason to suppose that adverse effects would be different for non-bacteraemic infections?

Round 2

Reviewer 4 Report

Thank you for making the revisions.